# MoRe4D: Joint 3D Motion Generation and Geometry Reconstruction for 4D Synthesis from a Single Image

## Abstract

Generating interactive, dynamic 4D scenes from *a single static image* remains a core challenge. Most existing *generate-then-reconstruct* and *reconstruct-then-generate* methods decouple geometry from motion, causing spatiotemporal inconsistencies and poor generalization. To overcome these limitations, we extend the reconstruct-then-generate framework to jointly couple **Mo**tion generation with geometric **Re**construction for **4D** Synthesis (**MoRe4D**). We first introduce TrajScene-60K, a large-scale dataset of 60,000 video samples with dense point trajectories, addressing the scarcity of high-quality 4D scene data. Based on this, we propose a diffusion-based 4D Scene Trajectory Generator (4D-STraG) to jointly generate geometrically consistent and motion-plausible 4D point trajectories. To leverage single-view priors, we design a depth-guided motion normalization strategy and a motion-aware module in 4D-STraG for effective geometry and dynamics integration. We then propose a 4D View Synthesis Module (4D-ViSM) to render videos with arbitrary camera trajectories from 4D point track representations. Extensive experiments show that MoRe4D generates high-quality 4D scenes with multi-view consistency and rich dynamic details from a single image.

## 1 Introduction

4D scene generation seeks to reconstruct comprehensive spatiotemporal representations capturing both explicit 3D geometry and complex temporal dynamics. Generating such dynamically rich 4D scenes from *a single static image* remains a fundamental challenge (Miao et al., 2025), as it requires recovering complete spatiotemporal information from inherently limited 2D observations. Achieving this capability would have a transformative impact on applications such as virtual reality (VR), augmented reality (AR) (Guo et al., 2023; Miao et al., 2024), and immersive content creation.

Although recent video generation models can produce realistic dynamic content, they generally lack an explicit understanding of 3D structure, leading to inconsistencies across views and failing to capture physically plausible motion. *Generate-then-reconstruct* methods (Wu et al., 2025; Yu et al., 2024a; Sun et al., 2024; Liu et al., 2025) use powerful video models to synthesize multi-view videos before reconstructing a 4D representation. This paradigm benefits from the high-fidelity and rich dynamics offered by existing video generation models. However, video models struggle to maintain strict geometric consistency across the generated views, leading to significant artifacts and structural collapse during the subsequent 3D reconstruction phase. Consequently, the alternative paradigm, *reconstruct-then-generate* (Zhao et al., 2023; Jiang et al., 2024; Bian et al., 2025; Jin et al., 2025; Ren et al., 2025b), has emerged as another promising direction. By first establishing a static 3D structure, this approach provides a robust geometric foundation for subsequent motion generation. Still, by decoupling static reconstruction from motion generation, they discard the rich dynamic potential latent in the source image. As a result, they are restricted to modeling physically plausible and externally constrained motions (e.g., swinging) and struggle to generate large-scale and self-initiated movements that originate from the scene or objects themselves.

To address these challenges, we propose an advanced *reconstruct-then-generate* framework that tightly couples motion generation with geometric reconstruction, as illustrated in Figure 1. To achieve joint modeling of motion and geometry, we propose to leverage dense point track as the

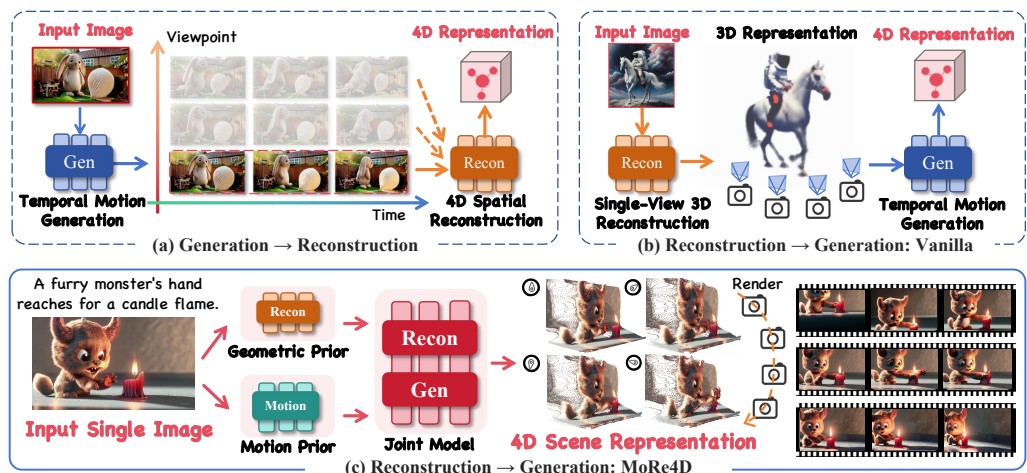

Figure 1: **MoRe4D for 4D synthesis from a single image.** Previous decoupled paradigms either suffer from geometric inconsistencies (*generate-then-reconstruct*) or are constrained by animating a pre-determined static geometry (vanilla *reconstruct-then-generate*). Our MoRe4D framework advances the latter paradigm by tightly coupling geometric modeling and motion generation, effectively achieving consistent 4D motion and geometry.

4D representation, which is directly predicted from the single input image with an integrated model. To facilitate training of the 3D motion generation model, we first built TrajScene-60K, a large-scale dataset of 60,000 samples with 4D point trajectories, tackling data scarcity of large-scale, high-quality 4D scene data with complex dynamics. We then introduce the *4D Scene Trajectory Generator* (4D-STraG), a novel diffusion model that, starting from an initial 3D reconstruction, uniquely enables the joint prediction of geometry and motion generation. Unlike prior works that treat these as separate steps, 4D-STraG operationalizes them within a unified denoising process, producing coherent 4D point trajectories whose motion is intrinsically consistent with the evolving 3D structure. To fully leverage priors from the input image, we further incorporate a depth-guided motion normalization method to enhance geometric awareness, and a Motion Perception Module (MPM) to leverage plausible motion priors. Finally, the generated 4D point cloud trajectories are rendered into high-fidelity dynamic videos from arbitrary novel viewpoints using our *4D View Synthesis Module* (4D-ViSM), completing the full pipeline from a single image to a coherent 4D scene.

Experimental results of both quantitative and qualitative comparisons demonstrate that our method consistently outperforms existing baselines, producing 4D scenes with more pronounced dynamics, stronger three-dimensional motion consistency, and superior numerical performance. In addition to overall comparisons, we conduct detailed visual ablation studies, which confirm the effectiveness of the key modules in our framework, highlighting how each component contributes to generating coherent and physically plausible 4D motion. These results collectively validate the capability of our approach to produce high-fidelity, multi-view consistent 4D scenes, paving the way for advanced VR/AR applications and immersive content creation. Our contributions can be summarized as:

- We construct TrajScene-60K, the first large-scale 4D scene dataset featuring point cloud trajectories, videos, and text to foster research in this domain.
- We propose 4D-STraG, a novel model to jointly perform reconstruction and generation from a single image, unifying geometry and dynamics in a single framework.
- We introduce innovative techniques, including depth-guided feature normalization and MPM, for stable and effective injection of geometric and motion priors.

## 2 RELATED WORK

**Video Generation.** The field aims to create temporally coherent and visually realistic dynamic visual content. From early VAE (Van Den Oord et al., 2017; Denton & Fergus, 2018; He et al., 2018;

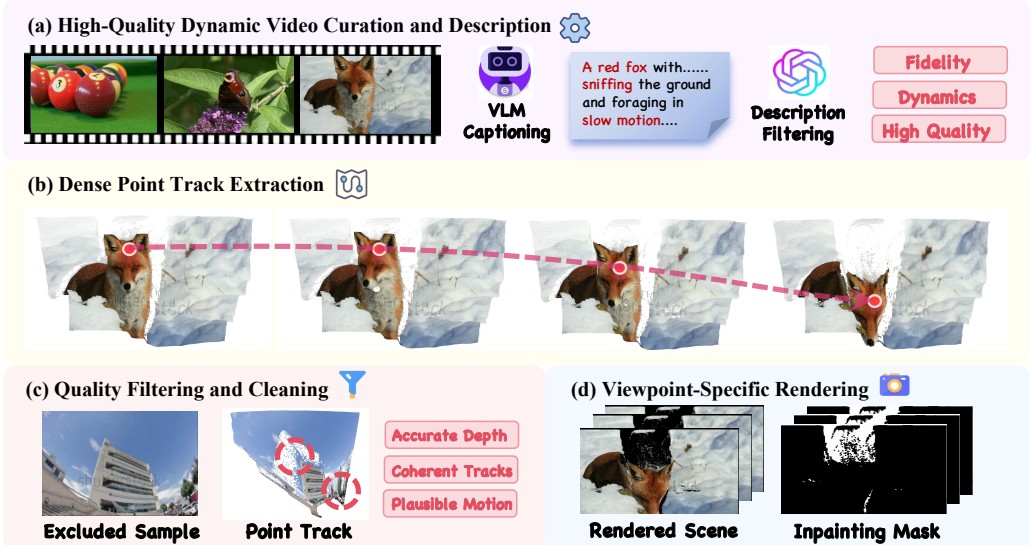

Figure 2: **TrajScene-60K curation pipeline.** We curate videos from WebVid-10M, filtered via VLMs for structured motion and countable entities. Dense 4D point tracks are extracted and refined via depth filtering and Gaussian Splatting, producing 60K high-quality 4D dynamic scenes.

Babaeizadeh et al., 2021; Bahmani et al., 2024) and GAN (Vondrick et al., 2016; Tulyakov et al., 2018; Clark et al., 2019) frameworks to modern large-scale diffusion models (Ho et al., 2022; Singer et al., 2023; Khachatryan et al., 2023; Wan et al., 2025; Team, 2024) trained on extensive video data, the realism and resolution of generated videos have been revolutionized. In addition to text control generation, researchers have introduced various guidance strategies for video generation, such as structure-based (Ma et al., 2024; Xing et al., 2024), image-based (Chen et al., 2023; Deng et al., 2024), and temporal controls (Shi et al., 2024; Wu et al., 2024b). However, these mainstream models are fundamentally limited as they operate in 2D pixel space, lacking explicit 3D scene modeling.

**Novel-View Synthesis (NVS).** 3D reconstruction-based methods reconstruct a 3D or 4D representation (Wu et al., 2024a; Zhou et al., 2023; Zhu et al., 2024; Charatan et al., 2024), ensuring strong geometric consistency but often requiring costly optimization and suffering from artifacts. Generation-based methods use pretrained video diffusion models conditioned on camera trajectories to synthesize novel views (Liu et al., 2024b; Yu et al., 2024b; Gao et al., 2024; Müller et al., 2024). While leveraging strong generative priors, they often exhibit inconsistencies and object drift under large viewpoint changes or complex scenes.

**4D Generation.** 4D generation aims to produce temporally evolving 3D representations from texts or sparse images. Existing methods largely follow two decoupled paradigms: *generate-then-reconstruct*, which first synthesizes videos and then reconstructs 4D (e.g., L4GM (Ren et al., 2025a), 4D-fy (Bahmani et al., 2024), 4Real (Yu et al., 2024a), CAT4D (Wu et al., 2025), DimensionX (Sun et al., 2024), Free4D (Liu et al., 2025)), and *reconstruct-then-generate*, which builds static 3D assets before animating them (e.g., Animate124 (Zhao et al., 2023), Animate3D (Jiang et al., 2024), GS-DiT (Bian et al., 2025), Jin et al. (2025), Gen3C(Ren et al., 2025b)). While effective, the former suffers from video–geometry mismatch, and the latter is often object-centric and limited in scene complexity. Our method advances the second line by introducing a joint framework that couples motion generation with geometric reconstruction, enabling consistent and coherent 4D synthesis.

## 3 DATASET CURATION – TRAJSCENE-60K

Training our 4D generation model requires a dataset encompassing three essential components: 4D point trajectories, viewpoint-specific observations, and semantic descriptions of 4D scenes. Given the scarcity of high-quality, large-scale annotated data, particularly for scene-level videos with significant motion and complex dynamics, we introduce TrajScene-60K, a comprehensive dataset designed to support robust 4D modeling of dynamic scenes.

As shown in Figure 2, we construct our dataset from the high-quality WebVid-10M corpus (Bain et al., 2021), extracting about 200,000 samples. To ensure data quality, we design an automated filtering pipeline with large language models: CogVLM2 (Hong et al., 2024) generates a descriptive caption $\mathcal{C}$ for each video, and DeepSeek-V3 (Liu et al., 2024a) evaluates these captions to retain videos with clearly countable entities undergoing self-initiated motion, while discarding those dominated by unstructured dynamics (e.g., crowd behaviors, wind-driven oscillations, or background jitter). This process yields samples with well-structured and quantifiable motion targets, enabling the learning of clear and interpretable dynamic representations.

To extract 4D dense point track data from videos, we employ the DELTA (Ngo et al., 2025) model. The model takes RGB video sequences $\mathcal{V} \in \mathbb{R}^{T \times H \times W \times 3}$ as input and utilizes monocular depth estimation methods to obtain depth maps $\mathcal{D} \in \mathbb{R}^{T \times H \times W}$, subsequently estimating occlusion-aware 4D trajectories $\mathcal{P} \in \mathbb{R}^{T \times H \times W \times 4}$. Each 4D vector $\mathbf{p}_{t,u,v} = (u_t, v_t, d_t, o_t)$ represents the 3D position and occlusion status in frame $t$ corresponding to the pixel located at $(u, v)$ in the first frame.

After obtaining 4D point cloud scene data, we perform quality filtering to remove samples with significant depth estimation errors, anomalous depth values, or missing values, which could ensure the accuracy and reliability of point track. This process yields 60,000 high-quality samples. Subsequently, we render these 4D scenes into videos from the original camera viewpoint using Gaussian Splatting (Kerbl et al., 2023). Inpainting masks are also generated for the void regions.

## 4 METHOD

### 4.1 PROBLEM DEFINITION

Given a single input image $\mathcal{I} \in \mathbb{R}^{H \times W \times 3}$ and its textual description $\mathcal{C}$, our goal is to reconstruct a physically plausible and temporally consistent 4D scene. We represent the scene as a point cloud sequence $\mathcal{P} \in \mathbb{R}^{T \times N \times 3}$, capturing both the 3D geometry and its motion trajectories of $N = H \times W$ points over $T$ frames. From this representation, we render dynamic scene videos $\mathcal{V} \in \mathbb{R}^{T \times H' \times W' \times 3}$ from arbitrary novel viewpoints or along arbitrary camera paths, thereby bridging the gap between a single static image and full multi-view 4D dynamic generation.

Unlike prior decoupled paradigms that suffer from error accumulation, our approach tightly integrates motion generation with geometric reconstruction. This ensures that the inferred dynamics are not only plausible but also intrinsically consistent with the evolving 3D structure. By co-optimizing structure and motion, we produce a more coherent and stable 4D representation.

To systematically achieve these objectives, we first design **4D Scene Trajectory Generator** (**4D-STraG**), a joint diffusion model that simultaneously reconstructs and generates spatiotemporal point trajectories from the initial image. Then we present in detail the **4D View Synthesis Module** (**4D-ViSM**), which leverages the reconstructed 4D representation to enable high-quality video generation under arbitrary camera motion. The overview illustration is shown in Figure 3.

### 4.2 **4D SCENE TRAJECTORY GENERATOR** (4D-STRAG)

To fully leverage the rich motion priors and structural understanding capabilities, we build upon the Wan 2.1 (Wan et al., 2025) architecture and finetune its spatiotemporal VAE and DiT modules separately. Below, we detail our module design and training scheme of 4D-STraG.

#### 4.2.1 POINT TRAJECTORY INITIALIZATION AND NORMALIZATION

To enhance training stability and ensure compatibility with generative model scales, we only use the 4D-STraG to predict relative motion $\mathbf{\Delta P}_t = \mathbf{P}_t - \mathbf{P}_0 = \{[\Delta x_t, \Delta y_t, \Delta z_t]\}$, where $t \in [0, T]$ and $\mathbf{P}_0$ denotes coordinates in the first frame. We further normalize $\mathbf{\Delta P}_t$ to avoid unlimited data value range. Based on the observation that a small 3D movement in a nearby object causes a large displacement on the 2D image plane, whereas the same movement in a distant object appears minuscule, we propose a *Depth-Guided Motion Normalization Strategy*. It normalizes the absolute motion of each point relative to its viewing frustum at the initial depth. By doing so, we transform raw motion into a scale-invariant representation, ensuring perceptual consistency across different distances and producing a more uniform data distribution for our generative model to learn from.

Figure 3: **Pipeline of MoRe4D.** Top: The 4D Scene Trajectory Generator (Sec. 4.2), a Diffusion Transformer, jointly generates geometry and motion. Bottom-Left: The Motion Perception Module (MPM) identifies potential motion regions and semantic structure from the input image. Bottom-Right: The 4D View Synthesis Module (Sec. 4.3) renders the output into novel-view videos.

Specifically, given focal lengths $f_x, f_y$ and image dimensions $W \times H$, we introduce the scaling factors $\alpha_x = f_x/W$ and $\alpha_y = f_y/H$. Geometrically, $z/\alpha_x$ and $z/\alpha_y$ correspond to the width and height of the viewing frustum at depth $z$, respectively. We normalize motion quantities by the viewing frustum size at depth $z = \mathbf{P}_0^{(z)}$:

$$\Delta \tilde{x}_t = \frac{\alpha_x \cdot \Delta x_t}{z}, \quad \Delta \tilde{y}_t = \frac{\alpha_y \cdot \Delta y_t}{z}, \quad \Delta \tilde{z}_t = \frac{\Delta z_t}{z}. \tag{1}$$

This depth-dependent normalization achieves scale invariance across different depth ranges, enabling our diffusion model to effectively learn motion patterns without being biased by the absolute spatial position of points. During inference, we use UniDepthv2 (Piccinelli et al., 2025) to estimate depth, ensuring consistency with the DELTA tracking model setup. The relative motion maps are de-normalized and fused with the initial point cloud to form a 4D scene representation.

### 4.2.2 MODEL ARCHITECTURE

**Model Pipeline.** During training, our model takes an image–caption pair as input and learns a diffusion model to predict pixel-level point trajectories across frames. To adapt the generative backbone, we first finetune a motion-sensitive VAE capable of handling trajectory signals. Specifically, the relative point displacement $\mathbf{\Delta P_t}$ is transformed into an RGB motion map by a lightweight Trajectory Encoder, where spatial movements are represented as color variations while the shape and appearance remains static as the first frame. Correspondingly, a Trajectory Decoder is appended after the VAE decoder, ensuring accurate recovery of point trajectories from the RGB motion map.

After finetuning the VAE, we adapt the Diffusion Transformer (DiT) to handle latents encoded from the RGB motion map. We explicitly inject strong geometric priors into the model, thereby enhancing its generative capability. Specifically, the depth information from the initial frame is encoded into a latent representation using the VAE encoder. This provides the model with robust structural priors and geometric cues, significantly improving its ability to reason about scene layout and object relationships. The image, noise, and depth latents are concatenated along the feature dimension to form the final input for the DiT:

$$z_{\text{combined}} = \text{Concat}(z_{\text{image}}, z_{\text{noise}}, z_{\text{depth}}). \tag{2}$$

The DiT model is trained using flow matching (Lipman et al., 2023), which learns deterministic flows from noise to data distributions by minimizing the error between predicted and true flow fields, enabling accurate modeling of pixel-level motion. The objective function is defined as:

$$\mathcal{L}_{\text{fm}} = \mathbb{E}_{t,x_0,x_1} \left[ |v_\theta(t, x_t) - (x_1 - x_0)|^2 \right], \tag{3}$$

where $x_0$ and $x_1$ represent the initial and target states, respectively, $x_t$ denotes the interpolated state at time $t$, and $v_\theta$ is the flow vector predicted by the model.

**Motion Perception Module (MPM).** A significant challenge lies in adapting the powerful motion priors learned by existing image-to-video diffusion models. These models excel at generating dynamic, moving scenes from a static image. Our objective, however, is to leverage this pretrained temporal knowledge for a distinct task: generating videos that are structurally static but exhibit dynamic color variations. This creates a core conflict, as the model must learn to translate its ingrained understanding of physical displacement into a new domain of temporal color evolution.

To effectively guide this adaptation and enable the model to identify which semantic regions within the static scene are candidates for these color dynamics, we introduce the Motion Perception Module. We employ OmniMAE (Girdhar et al., 2023), pretrained on image-video joint data, as our Motion Feature Extractor to obtain motion-aware patch-level features $\mathbf{S}$ from static images. To embed motion information into the diffusion process, we introduce *Motion-aware Adaptive Normalization* (MAdaNorm), which performs fine-grained feature modulation in each DiT layer. Specifically, after spatial alignment of motion features $\mathbf{S}$ by resizing them to match the DiT token sequence length, token-wise adaptive parameters are generated through linear layers. For intermediate features $\mathbf{F}_t^i \in \mathbb{R}^{N \times d}$ in the $i$-th DiT block, the modulation process is as follows:

$$\boldsymbol{\alpha}_1, \boldsymbol{\alpha}_2, \boldsymbol{\beta}_1, \boldsymbol{\beta}_2 = \text{Linear}(\mathbf{S}), \tag{4}$$

$$\mathbf{F}' = \text{Attn}\left(\boldsymbol{\gamma_1}\boldsymbol{\alpha}_1 \odot \text{LN}(\mathbf{F}_t^i) + \boldsymbol{\gamma_1}\boldsymbol{\beta}_1\right), \quad \mathbf{F}'' = \text{MLP}\left(\boldsymbol{\gamma_2}\boldsymbol{\alpha}_2 \odot \text{LN}(\mathbf{F}') + \boldsymbol{\gamma_2}\boldsymbol{\beta}_2\right), \tag{5}$$

where $\boldsymbol{\alpha}_1, \boldsymbol{\alpha}_2, \boldsymbol{\beta}_1, \boldsymbol{\beta}_2 \in \mathbb{R}^{N \times d}$ are token-wise scaling and bias parameters, $\boldsymbol{\gamma_1}, \boldsymbol{\gamma_2} \in \mathbb{R}^d$ is a learnable global gating coefficient, and $\odot$ denotes token-wise multiplication. Experimental results demonstrate that MPM significantly improves the fidelity of motion trajectory modeling and enhances the spatio-temporal consistency of generated 4D content.

### 4.3 4D VIEW SYNTHESIS MODULE (4D-VISM)

After obtaining a dense 4D point cloud representation, we propose the 4D-ViSM to achieve novel view video synthesis along arbitrary camera trajectories. The original point cloud may not fully cover image regions in novel views, which results in holes in the rendered output. We thereby leverage generative models to complete these missing regions, ensuring visual coherence and plausibility.

Considering excellent performance of Wan2.1 (Wan et al., 2025) in video generation tasks, we also choose to finetune this model to build our 4D-ViSM. The training data also come from our TrajScene-60K dataset, including rendered videos, corresponding occlusion masks, ground truth videos, and captions. During training, we follow Wan2.1 mask processing strategy, setting the mask value to 0.5 for regions without projected points in each frame. Through finetuning, our model achieves high-quality and visually consistent novel view video synthesis.

## 5 EXPERIMENTS

### 5.1 EXPERIMENTAL SETUP

**Baseline Methods.** We compare it with leading 4D generation methods: 4Real (Yu et al., 2024a), which adapts a pre-trained 3D model on synthetic data; GenXD (Zhao et al., 2025), disentangling camera and object motion; Gen3C (Ren et al., 2025b), using 3D caching for camera control and consistency; DimensionX (Sun et al., 2024), employing decoupled spatio-temporal video diffusion; and Free4D (Liu et al., 2025), a training-free approach distilling foundational models for single-image 4D generation. Among them, 4Real, GenXD, DimensionX, and Free4D follow a *generate-then-reconstruct* pipeline, while Gen3C adopts a *reconstruct-then-generate* strategy. We later provide a full qualitative and quantitative assessment in Section 5.2 and Section 5.3.

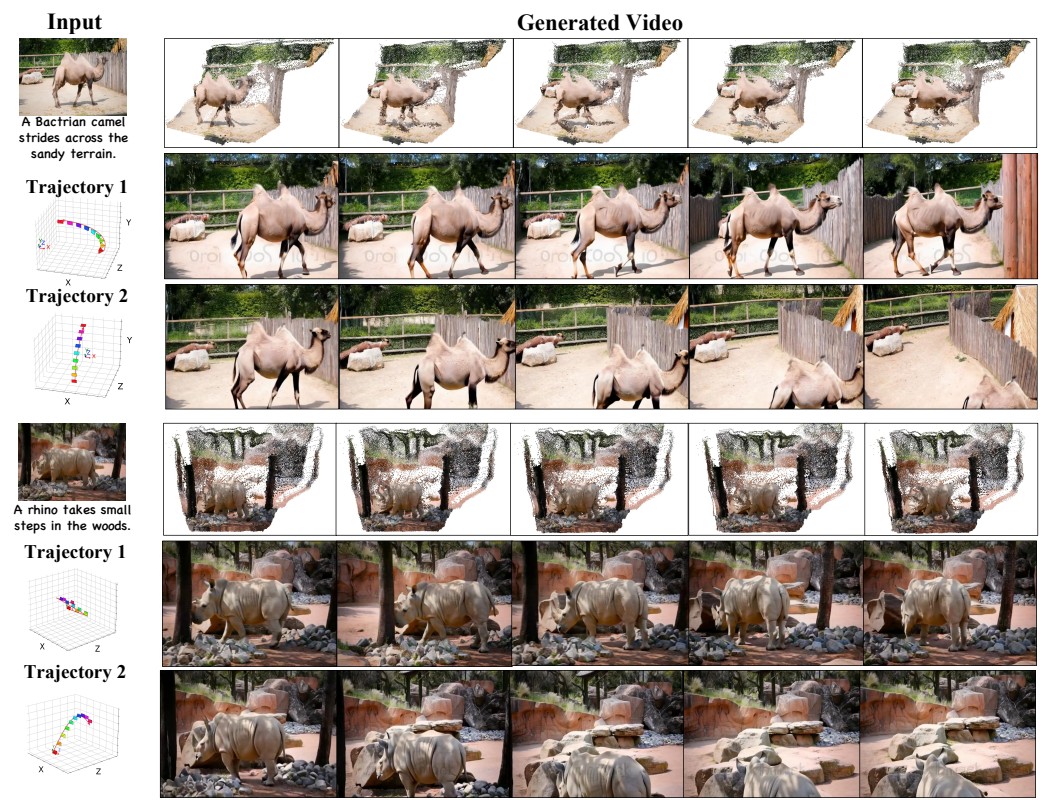

Figure 4: **Qualitative results of our model.** We visualize the generated results for two samples. For each, the first row shows the 4D point cloud generated by our 4D-STraG. The second and third rows show the videos rendered by our 4D-ViSM under two distinct, user-defined camera trajectories.

**Implementation Details.** Our model was trained on our TrajScene-60K dataset. The model generates videos at a resolution of $512 \times 368$ with a length of 49 frames. The 4D-STraG module, we performed full-parameter training based on Wan2.1-14B. The trajectory encoder and decoder are both shallow ResNets. We first trained its tracking components and VAE decoder for 5k steps, then its DiT for 2k steps using OmniMAE features for motion conditioning. The 4D-ViSM module fine-tuned Wan2.1-14B with LoRA for 10k steps. We used the AdamW optimizer with a learning rate of $2 \times 10^{-5}$. All experiments ran on four NVIDIA H20 GPUs.

## 5.2 QUALITATIVE RESULTS

For qualitative validation of our model effectiveness, we conduct comparative analyses in Fig. 4 and Fig. 5. Fig. 4 presents two representative scenes, with the first row displaying the point cloud outputs of 4D-STraG. It can be observed that the 4D point clouds maintain structural consistency and exhibit reasonable motion over time, with high detail completeness. For each scene, we define two camera trajectories and perform rendering using 4D-ViSM. The results demonstrate that the rendered images not only preserve the geometric accuracy of the original structure but also effectively align with the camera trajectories, reflecting strong visual consistency. Fig. 5 further compares the visual outcomes of several existing methods, including 4Real, DimensionX, Gen3C, and Free4D. In comparison to these approaches, our method generates more diverse and realistic motion, validating the advantage of the "joint reconstruction and generation" strategy in ensuring both structural rationality and motion plausibility for 4D generation.

## 5.3 QUANTITATIVE RESULTS

We further validated our model effectiveness through quantitative experiments on VBench (Huang et al., 2024), a comprehensive benchmark assessing video generation across six dimensions: Subject

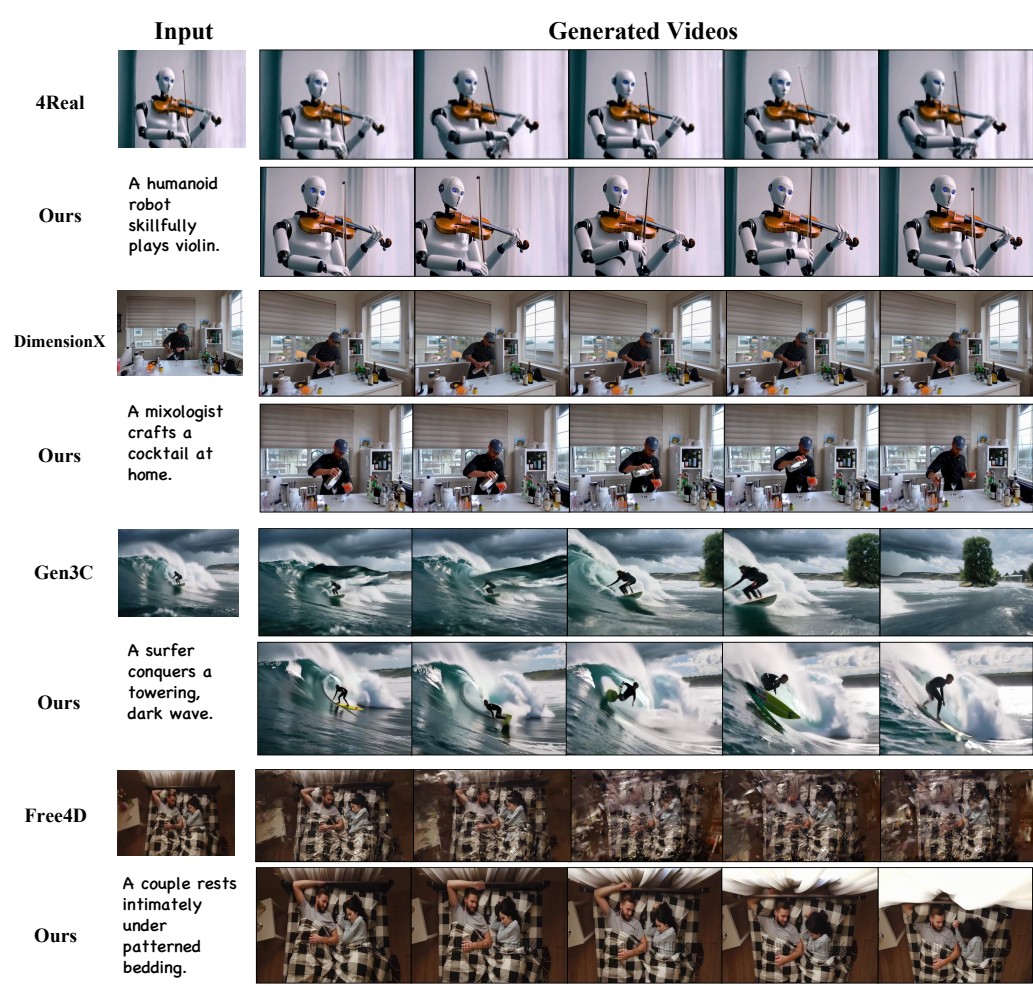

Figure 5: **Qualitative comparison with baselines methods.** For each sample, the first row shows the baseline results while the second row presents our MoRe4D results. The first column displays the input image and text prompt.

and Background Consistency, Temporal Smoothness, Dynamics, Aesthetic Quality, and Imaging Quality. As shown in Table 1, we structured our comparisons into three groups based on model availability and input modalities. First, we compared against the closed-source 4Real by inferring from the first frame of its official 4D generation videos. Our model demonstrated significantly better Dynamics, Aesthetic, and Imaging Quality. For the second and third groups, we used 100 samples from WebVid-10M to evaluate performance in complex scenarios. We benchmarked against single-image 3D reconstruction models (GenXD and the open-source S-Director from DimensionX) by applying a fixed 90-degree rotation to their static outputs, where MoRe4D surpassed them in consistency and visual quality. Finally, when compared with other 4D generation models under more challenging camera trajectories, MoRe4D again achieved state-of-the-art results on multiple visual metrics, with a pronounced lead in Aesthetic and Imaging Quality.

## 5.4 ABLATION STUDIES

**Point Normalization Strategy.** Row 1-2 in Figure 6 demonstrates the effectiveness of our depth-guided motion normalization. A baseline normalization using min-max coordinates yields unstable point clouds, failing to preserve the object structure or model plausible motion. This instability is particularly pronounced for scenes with large depth variations, where it causes exaggerated movements for nearby objects. Our depth-guided approach effectively resolves this issue.

Table 1: **Quantitative comparison on VBench.** Higher values are better. The best results in each comparison group are marked in **bold**.

| Exp. No. | Model / Metrics | Subject Consistency | Background Consistency | Motion Smoothness | Dynamic Degree | Aesthetic Quality | Imaging Quality |
|---|---|---|---|---|---|---|---|
| **I** | 4Real (Yu et al., 2024a) | **0.9329** | **0.9709** | 0.9664 | 0.7708 | 0.4938 | 0.5095 |
| | MoRe4D(Ours) | 0.8752 | 0.9364 | **0.9682** | **1.0000** | **0.5613** | **0.6230** |
| **II** | GenXD (Zhao et al., 2025) | 0.8042 | 0.8789 | 0.9030 | **1.0000** | 0.4077 | 0.5209 |
| | DimensionX (Sun et al., 2024) | 0.7553 | 0.8481 | **0.9827** | **1.0000** | 0.4634 | 0.5545 |
| | MoRe4D(Ours) | **0.8241** | **0.9044** | 0.9760 | 0.9500 | **0.4820** | **0.5828** |
| **III** | Free4D (Liu et al., 2025) | 0.7899 | 0.8883 | 0.9797 | **1.0000** | 0.3607 | 0.3562 |
| | Gen3C (Ren et al., 2025b) | 0.8112 | 0.8871 | **0.9845** | 0.9940 | 0.3812 | 0.4814 |
| | MoRe4D(Ours) | **0.8339** | **0.9065** | 0.9773 | 0.9000 | **0.4820** | **0.5939** |

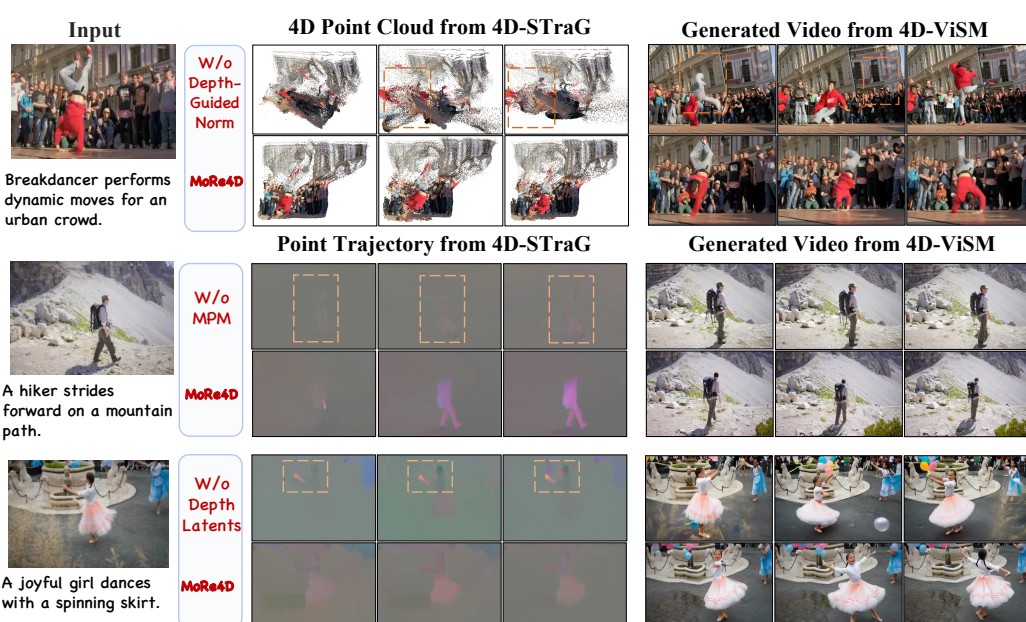

Figure 6: **Ablation studies on normalization methods and module components.** (Rows 1-2) Depth-guided motion normalization stabilizes 4D point cloud generation. (Rows 3-6) Removing the MPM module reduces motion magnitude while excluding depth guidance breaks structural motion consistency, validating our design choices.

**Module Ablations.** Further ablations on our 4D-STraG module are shown in row 3-6 in Figure 6. Removing the Motion Perception Module (MPM) severely diminishes the predicted motion magnitude, highlighting its necessity for inferring dynamic 4D representations from a single image. Furthermore, when depth is excluded as geometric guidance, the model struggles to maintain motion consistency across different parts of the same object. This confirms the critical role of depth in ensuring structurally coherent point trajectories.

# 6 CONCLUSION

We proposed MoRe4D, a unified framework for single-image 4D generation that tightly couples motion and geometry, overcoming the inherent weaknesses of decoupled paradigms. To support training, we constructed TrajScene-60K, a large-scale dataset with dense 4D trajectories, and introduced the 4D-STraG diffusion model with depth-guided motion normalization and motion perception priors for joint inference. Experiments show that MoRe4D achieves superior geometric consistency, dynamic realism, and visual fidelity compared to existing approaches. This work establishes a new technical pathway for 4D synthesis from minimal input, and points toward future exploration of more universal dynamic priors and lightweight 4D representations for practical deployment.

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

# A APPENDIX

## A.1 DATASET CURATION

**Video Filtering and Annotation.** We initiated our process with approximately 200,000 video candidates from the WebVid-10M dataset. To automate the selection of content suitable for 4D dynamic scene modeling, we implemented a two-stage pipeline. First, we employed the multimodal large language model CogVLM2 to generate a detailed English caption for each video. Subsequently, these captions were fed into the DeepSeek-V3 model, which used a carefully designed prompt to assess content suitability. The core evaluation criteria were: (1) the presence of one or more clearly countable entities, and (2) the exhibition of self-initiated, non-rigid, or articulated motion, as opposed to random movements driven by external forces (e.g., wind, water) or dominant camera motion. This process effectively filtered out videos featuring unstructured dynamics, such as crowd movements, water ripples, or swaying foliage. Our prompt is shown in Figure 7.

> Identify whether the caption describes countable moving entities. An entity is considered moving if:
> 1) it moves by itself,
> 2) part of it moves, or
> 3) it appears to move due to camera movement.
> However, uncountable phenomena (like waterfalls, rivers, particles, light beams, text animations, crowds too numerous to count) do NOT qualify as countable moving entities.
>
> Let me think through this step by step:
> 1. What entities are described in the caption?
> 2. Is any entity or part of an entity moving (or does it appear to move due to camera movement)?
> 3. Can these moving entities be counted, or are they uncountable phenomena?
> 4. Based on this analysis, determine if there are countable moving entities.
>
> Your conclusion: After analyzing the caption step by step, I determine the answer is {result: <flag>}, where <flag> is true or false.

Figure 7: **Our input prompt.** The standardized prompt used to query the model for identifying and counting self-initiated, articulated motion.

**4D Trajectory Quality Control.** After extracting the raw 4D trajectories, we applied a strict quality filtering process to eliminate samples compromised by depth estimation failures or extreme motion. The specific criteria were as follows: (1) We removed samples where a significant portion of point cloud trajectories contained invalid or anomalous depth values (e.g., near-infinite or zero) at any timestep. (2) We discarded samples exhibiting an excessively large standard deviation in scene depth, which indicates potential errors in the global depth estimation. (3) We performed a scale consistency check. Since uniformly scaling a point cloud should yield an identical rendering from the original camera perspective, we removed samples where this transformation resulted in significant visual changes, flagging them as geometrically inconsistent. This comprehensive filtering pipeline yielded the final 60,000 high-quality samples that constitute the TrajScene-60K dataset.

## A.2 MODEL ARCHITECTURE DETAILS

### A.2.1 DETAILS OF MOTION-SENSITIVE VAE ARCHITECTURE

Before training the DiT in 4D-STraG module, we finetune a specialized, motion-sensitive VAE to effectively adapt our generative backbone for trajectory synthesis. This VAE is designed to process

Figure 8: **Qualitative results of Motion-Sensitive VAE.**

and reconstruct trajectory information encoded as RGB-like motion maps. Inspired by Geo4D (Jiang et al., 2025), The architecture of both the VAE encoder and decoder is intentionally kept shallow to preserve fine-grained motion details. The Trajectory Encoder and Decoder that process these motion maps are constructed as shallow ResNets (He et al., 2016). This minimalist design ensures that the VAE learns a compact and efficient latent space for motion patterns without aggressively downsampling the spatial features, which is critical for the precise reconstruction of trajectories by the Trajectory Decoder. When finetuning Motion-Sensitive VAE, we trained the Trajectory Encoder, Trajectory Decoder, and the VAE Decoder, while freezing the VAE Encoder. This approach allows the model to fully adapt to the motion input while preserving the integrity of the pre-trained visual representations.

### A.2.2 DETAILS OF 4D-ViSM ARCHITECTURE

Our 4D-ViSM model adapts a pre-trained video Diffusion Transformer (DiT) for dynamic video inpainting, specifically to fill holes in novel-view videos rendered from our 4D representation. During fine-tuning, the model is conditioned on the incomplete video. For each step, the rendered video with holes is encoded into a latent representation $z_{\text{rendered}}$, and its binary occlusion mask is downsampled to $m_{\text{latent}}$. The denoising network's input is a channel-wise concatenation of the noisy latent $z_t$, $z_{\text{rendered}}$, and $m_{\text{latent}}$, formulated as

$$z = \text{concat}([z_t, z_{\text{rendered}}, m_{\text{latent}}]).$$

This explicitly provides the model with the known visual context and the missing regions, enabling coherent video completion.

### A.2.3 INFERENCE PIPELINE

Given a single image and a text prompt, our model first acquires geometric information by estimating depth through the 4D-STraG module. Specifically, we utilize the UniDepthv2 (Piccinelli et al., 2025) model to infer depth information, ensuring consistency with the estimation method used for the DELTA tracking model during training. The VAE then encodes the image and depth map into a latent space, while MPM extracts motion features. Subsequently, DiT generates latent representations, which are decoded into relative motion latents. These latents are de-normalized by reversing our depth-guided motion normalization strategy and then fused with the initial point cloud's spatial coordinates to construct a 4D scene representation. Finally, this representation allows for rendering from arbitrary camera poses, and our 4D-ViSM synthesizes a spatio-temporally consistent 4D video aligned with the desired camera trajectory.

### A.3 ACKNOWLEDGMENT OF LLM USAGE

A large language model (LLM) was employed for minor language editing, including grammar checking and sentence rephrasing, to enhance the clarity and readability of the manuscript. Additionally, as mentioned in Section 3, the LLM (DeepSeek-v3) was used for filtering the dataset. While the LLM contributed to data processing, it did not influence the research ideation, methodology, or scientific conclusions.

### A.4 MORE QUALITATIVE RESULTS OF MOTION-SENSITIVE VAE

As shown in Figure 8, we present the reconstruction results using the fine-tuned VAE on a subset of samples during inference. The reconstructed points are rendered from the original camera viewpoint, demonstrating that they effectively preserve the structural and motion consistency with the input. More specifically, the model exhibits robust representation learning capabilities, enabling effective extraction of latent space representations that accurately capture both geometric and dynamic properties of the observed scenes.

