# OpenReview forum: "MoRe4D: Joint 3D Motion Generation and Geometry Reconstruction for 4D Synthesis from a Single Image"
_ICLR.cc/2026/Conference — ICLR 2026 Conference Withdrawn Submission_

### Official Review · Reviewer_KDft · 2025-10-30

**Soundness:** 1
**Presentation:** 3
**Contribution:** 2
**Rating:** 2
**Confidence:** 4

**Summary:**

The paper proposes MoRe4D, a "reconstruct-then-generate" framework that jointly couples 3D geometry with motion for single-image to 4D scene synthesis. The paper puts forth 3 main contributions:
(1) TrajScene-60K, a curated dataset of ~60k videos with dense 4D point trajectories derived from WebVid using VLM-based filtering, monocular depth, and tracking.
(2) 4D-STraG, a diffusion/flow-matching model that predicts relative, depth-normalized point motions from a single image, using a Motion Perception Module.
(3) 4D-ViSM, a video diffusion inpainting stage to render novel-view videos from the generated 4D point trajectories, filling holes where projections are sparse.

Experiments emphasize qualitative results and VBench scores, with ablations for depth-guided motion normalization and the MPM.

**Strengths:**

- The paper is easy to follow. The roles of 4D-STraG vs. 4D-ViSM are well separated and explained.
- The proposed dataset seems highly beneficial for 3D/4D fields, as well as motion tracking fields.

**Weaknesses:**

- The proposed pipeline seems to be a cleverly integrated vision system that combines many different pretrained models. It is unclear what the contributions here are other than smart system integration. It is also unclear if any of this transfers if we use other pretrained priors.
- Evaluation does not target true 4D consistency/geometry. Given the paper’s central claim (joint geometry–motion), this is a mismatch: the method should be measured on whether the generated 4D is geometrically coherent across novel views and time. With TrajScene-60K in hand (pseudo-GT tracks and depths), it’s feasible to report 2D endpoint error (EPE) of tracks, occlusion accuracy, 3D Chamfer/EPE3D against pseudo-GT or self-consistency checks across rendered views. There are also other synthetic datasets that can be employed to measure this.
- The paper does not report on established dynamic NVS/4D datasets (even with caveats), nor does it propose a public evaluation split for TrajScene-60K with released annotations to allow reproducible comparisons. Without public release/eval code, the benchmark story is weak.
- Baseline evaluations and experiment design is very confusing. Different models are benchmarked in different ways, without a clear explanation on why these evaluations make sense. To me it seems like the authors are cherry-picking evaluations.
- Ablations are qualitative only. Not even VBench evaluations are included.

Minor points:
- Feature-wise conditioning reminds me of FiLM/AdaIN/T2I-Adapter style conditioning. They should be credited.
- Depth-scaled normalization: commonly used in geometric vision models and scale-invariant depth or flow models. Should also be cited.

**Questions:**

In additional to the weakness section above:
- How often does 4D-ViSM “hallucinate” content outside the projected points, and how does that affect multi-view consistency over time?  - What are the dominant failure modes?
- How sensitive is performance to depth noise?

**Details Of Ethics Concerns:**

The dataset relies on WebVid plus monocular depth/tracking and LLM/VLM filters, but the paper doesn’t clarify the bias induced by LLM/VLM filtering.

---

### Official Review · Reviewer_LyB9 · 2025-11-01

**Soundness:** 3
**Presentation:** 2
**Contribution:** 2
**Rating:** 4
**Confidence:** 4

**Summary:**

This paper presents MoRe4D, a novel framework for joint 3D motion generation and geometry reconstruction to synthesize a dynamic 4D scene from a single static image. The core innovation is to move beyond the traditional decoupled "generate-then-reconstruct" (GtR) or "reconstruct-then-generate" (RtG) pipelines, which often lead to spatiotemporal inconsistencies. MoRe4D introduces a 4D Scene Trajectory Generator (4D-STraG), which is a spatiotemporal diffusion model (finetuned from Wan 2.1) trained to predict a dense 4D point trajectory ($\Delta P_t$) coupled with an initial 3D point cloud ($P_0$). To achieve this geometric awareness, the method incorporates a Depth-Guided Motion Normalization and a Motion Perception Module (MPM) with MAdaNorm to inject motion-aware features and enforce consistency. The method demonstrates superior performance in generating coherent 4D dynamic point clouds and views, validated on a newly introduced, large-scale TrajScene-60K dataset.

**Strengths:**

1. The introduction of the TrajScene-60K dataset, a large-scale collection of 4D trajectories coupled with 3D geometry, is a valuable contribution that will facilitate future research in this domain.
	2. generated dynamic scenes and resulting dynamic point clouds exhibit high fidelity, fine-grained details, and superior spatiotemporal consistency compared to baseline methods, particularly those relying on sequential reconstruction.

**Weaknesses:**

1. The image-to-motion generation ability comes from Wan-I2V. What is the difference with the pipeline that first generating a video then conducting 4D reconstruction with methods like TTT3R or MASt3R? This pipeline could also get a dynamic point cloud as output, expressing the unnecessary need to do the complex joint trajectory generation proposed by the authors.

2. Regarding MPM, integrating an external feature extractor (OmniMAE) with a conventional conditional injection mechanism has few genuinely novel technical contributions and could be considered a relatively simple way to inject conditional information into the diffusion transformer architecture.

3. Difference with 4DNeX [1]: The paper needs to clearly articulate the technical superiority of its joint motion/geometry prediction architecture (MoRe4D) compared to other feed-forward models like 4DNeX, which also fine-tunes a pretrained video diffusion model to predict a unified 6D (RGB+XYZ) representation from a single image. The claimed benefit of coupling motion and geometry must be empirically shown to surpass other unified, feed-forward approaches.

4. (minor, misclassification) In L148, 4dfy first conforms the "reconstruct -then-generate" instead of the previous one paradigm. It first generate canonical NeRF with SDS. Instead, SV4D [2], EG4D [3] conforms the "reconstruct -then-generate" scheme.


[1] 4DNeX: Feed-Forward 4D Generative Modeling Made Easy

[2] SV4D: Dynamic 3D Content Generation with Multi-Frame and Multi-View Consistency, ICLR 2025

[3] EG4D: Explicit Generation of 4D Object without Score Distillation, ICLR 2025

**Questions:**

see weakness

---

### Official Review · Reviewer_nPQj · 2025-11-01

**Soundness:** 2
**Presentation:** 3
**Contribution:** 3
**Rating:** 6
**Confidence:** 4

**Summary:**

The paper propose to generate geometry and motion jointly for 4D synthesis from a single image. Specifically, the authors first construct a large 4D dataset with 4D annotations from a pre-trained 3D point trajectory method. Then, they finetune a video diffusion model to predict point cloud sequence conditioned on the input image and depth. They also train a 4D view synthesis module to predict video from the projection of the generated point cloud sequence. Extensive experiments on public and private dataset demonstrate the effectiveness of the proposed method.

**Strengths:**

1)The motivation of jointly predicting geometry and motion makes sense.

2)The paper conduct extensive experiments.

3)Generated results seems great.

**Weaknesses:**

1) Lack of evaluation for 4D consistency: it is suggested to reconstruct the 4D scene using nerf or gaussian to better evaluate the 4D consistency of the proposed method as well as other methods

2) The authors are suggested to validate their joint prediction design by comparing with video gen from Wan2.1 + DELTA point tracking + 4D-ViSM result (as a generation-reconstruction method, close to the paper setting), since they use DELTA for video annotation

3) It is recommended to incorporate point trajectories in conjunction with point cloud rendering in 4D ViSM

**Questions:**

See weaknessnes

---

### Official Review · Reviewer_CNBJ · 2025-11-01

**Soundness:** 3
**Presentation:** 3
**Contribution:** 3
**Rating:** 6
**Confidence:** 4

**Summary:**

This paper proposes a  4D scene, i.e., temporally dynamic 3D scene, synthesis method, which adopts reconstruct-then-generate strategy.  It first generates 4D point trajectories conditioned on input image and text, and then renders the 4D point trajectories with 3D Gaussian splatting as the initial video.  Finally, the initial video is used as the guidance for the final video generation. Since  this method reconstructs 4D point trajectories as the underlying representation of the 4D scene, it supports the editing of camera trajectories during video generation.

**Strengths:**

1.  The 4D point trajectory representation is novel, and it is effective to serve as an intermediate representation to improve geometric consistency in video generation.
2.  This paper constructs TrajScene-60K, with 60,000 video samples with dense point trajectories. This dataset is valuable to the video processing research.
3.  the point trajectory normalization step is effective to facilitate the training of VAE for 4D point trajectories.

**Weaknesses:**

The generated video is relatively short, and in its current form, the proposed method would likely struggle to handle the appearance or disappearance of objects mid-sequence.

**Questions:**

Suppose a video sequence recorded for a man takes out his mobile phone from his pocket at mid-sequence, but the mobile phone is not visible in the first frame because it is still in the pocket,  how do you construct the 4D point trajectories for this sequence?

---

### Note · Authors · 2025-11-14

I have read and agree with the venue's withdrawal policy on behalf of myself and my co-authors.